# The Impact of Iron on Cancer-Related Immune Functions in Oncology: Molecular Mechanisms and Clinical Evidence

**DOI:** 10.3390/cancers16244156

**Published:** 2024-12-13

**Authors:** Omar Badran, Idan Cohen, Gil Bar-Sela

**Affiliations:** 1Department of Oncology, Emek Medical Center, Afula 1834111, Israel; hfomerbad@clalit.org.il (O.B.); idan5161@gmail.com (I.C.); 2Technion Integrated Cancer Center, Faculty of Medicine, Technion, Haifa 3525422, Israel

**Keywords:** iron deficiency, iron supplementation, immunotherapy, oncology patients, immune system, tumor microenvironment

## Abstract

Iron is essential for many bodily functions, including energy production and immune defense. However, in cancer patients, both low and high iron levels can negatively impact disease progression and treatment effectiveness. Iron deficiency can weaken the immune system, reducing the effectiveness of therapies that rely on immune responses. On the other hand, excess iron can promote tumor growth and create an environment that protects cancer cells from treatments. This review explores how managing iron levels in cancer patients may improve treatment outcomes, aiming to guide future research and help develop better strategies for personalized cancer care.

## 1. Introduction

Iron is essential for various biological processes, including oxygen transport, DNA synthesis, and cellular respiration [1]. Its critical role in hemoglobin and myoglobin enables the efficient transport and storage of oxygen, supporting the metabolic demands of tissues throughout the body [2]. Additionally, iron serves as a cofactor for numerous enzymes involved in vital cellular functions such as energy production and DNA repair [3]. However, iron’s reactive nature can pose significant risks despite its indispensable role, particularly in cancer [4].

Cancer cells deregulate iron homeostasis to support their growth, proliferation, and phenotypic traits such as invasiveness and aggressiveness [5]. This deregulation is characterized by an increased labile iron pool (LIP), upregulation of transferrin receptor (TfR) expression on the cytoplasmic membrane, and specific modulation of iron regulatory proteins (IRPs), particularly IRP2, which serves as a key regulator of iron metabolism in tumor cells [6,7,8,9]. These mechanisms create an iron-rich microenvironment within cancer cells, promoting their malignant behavior.

In contrast, cancer patients often experience systemic iron deficiency and anemia due to factors such as chronic inflammation, malnutrition, and blood loss [10]. Despite this systemic deficiency, the localized iron overload within tumor cells significantly drives tumor progression and influences systemic iron metabolism [11]. This duality highlights the complex interplay between iron metabolism in cancer cells and systemic iron homeostasis in patients. In oncology, the management of iron presents a complex challenge due to its dualistic nature. On the one hand, iron deficiency is a common issue among cancer patients [12], often exacerbated by the disease itself [13]. This deficiency can lead to anemia, significantly impairing patients’ quality of life, reducing physical performance, and lowering their capacity to tolerate treatments like chemotherapy and radiotherapy [14]. Moreover, anemia resulting from iron deficiency may worsen the overall prognosis by limiting oxygen supply to tissues, thus reducing the efficacy of therapies, such as radiation, that depend on oxygen for maximum effectiveness [15]. Iron deficiency can significantly weaken immune function, hindering the body’s ability to mount an effective immune response against the tumor [16]. This can negatively impact immunotherapies such as immune checkpoint inhibitors (ICIs) and chimeric antigen receptor (CAR)-T cell therapies, which rely on a robust immune system to be effective.

On the other hand, iron overload, which may arise from frequent blood transfusions, overuse of iron supplements, or genetic conditions such as hereditary hemochromatosis (HH) [17,18], can facilitate the production of reactive oxygen species (ROS) through the Fenton reaction [19]. This results in oxidative stress, DNA damage, and the creation of a pro-tumorigenic environment that promotes cancer cell proliferation, invasion, and metastasis [20]. Excess iron can also have toxic effects on multiple organs, including the liver, heart, and endocrine system [18,19,20,21]. Iron overload in the liver can damage and impair its function [22], which is concerning as it plays a critical role in metabolizing and processing many chemotherapeutic agents [23]. Impaired liver function due to iron overload may hinder the body’s ability to metabolize these drugs effectively, potentially reducing the efficacy of chemotherapy and immunotherapy [23].

Recent advancements in cancer treatment have further highlighted the importance of iron metabolism in therapeutic outcomes. For instance, anti-cancer immunotherapies, like ICIs, depend on a fully functional immune system to target and eliminate cancer cells [24]. However, iron deficiency can impair the immune response, particularly in critical immune cells such as T-cells, macrophages, and natural killer (NK) cells, which are crucial for the success of immunotherapy [25]. Some studies have shown that iron supplementation can positively influence the efficacy of immunotherapies [26]. Conversely, iron overload can disrupt the balance and function of T-lymphocyte subsets, altering their surface markers and distribution within the immune system, thereby weakening the immune response against tumors [27]. Lymphocytes with limited iron storage capacity may also become more vulnerable to damage due to iron overload, further contributing to immune dysfunction in patients with high iron levels [28].

Given iron’s critical role in cancer progression and immune function, this review explores the intricate interplay between iron metabolism, cancer biology, and therapeutic outcomes. We have collated and analyzed the current research findings to provide a comprehensive understanding of how variations in iron levels can influence cancer treatment efficacy. By examining the latest studies and emerging evidence, we illustrate how iron levels impact tumor growth, the tumor microenvironment (TME), and the effectiveness of therapies. Furthermore, we propose several strategies for optimizing iron management in oncology patients to enhance therapeutic efficacy. Ultimately, this review emphasizes the importance of understanding and managing iron metabolism in cancer care, encouraging a more holistic approach that considers iron’s role in both the tumor and the immune system.

## 2. Iron Deficiency in Cancer Patients

Iron deficiency in cancer patients is a significant complication with a multifactorial etiology [29]. Its prevalence ranges between 30% and 60%, depending on the type of cancer and treatment modalities involved [30]. The causes of iron deficiency are often related to chronic blood loss, inadequate nutritional intake, side effects of treatments, and malabsorption [29,31] (Figure 1). One of the main drivers of this deficiency is the chronic inflammatory response associated with cancer itself [31].

Functional iron deficiency (FID) is a common condition in cancer patients despite adequate iron stores in the body [32]. This anemia arises from iron sequestration, which renders it unavailable for critical processes like erythropoiesis, reducing red blood cell production [33]. FID significantly impacts cancer patients, contributing to fatigue, diminished quality of life, and the reduced efficacy of cancer treatments [34,35].

The regulation of iron in the body is primarily governed by hepcidin, a hormone produced in the liver [36]. Hepcidin is central in maintaining systemic iron homeostasis by inhibiting iron absorption in the gastrointestinal tract and reducing iron release from macrophages, hepatocytes, and enterocytes [37]. In the duodenum, dietary iron is absorbed through DMT1 (divalent metal transporter 1), a key protein responsible for transporting non-heme iron from the intestinal lumen into enterocytes [38]. Once inside the enterocytes, iron can be stored as ferritin or exported into the bloodstream via ferroportin (SLC40A1), the only known iron exporter in mammalian cells [39]. Hepcidin binds to ferroportin, triggering its internalization and degradation, thereby reducing iron transport into the bloodstream [40]. This tightly regulated process prevents iron overload but, when dysregulated, contributes to conditions such as FID. In cancer patients, elevated hepcidin levels disrupt this balance, reducing iron availability for erythropoiesis and other critical functions, further impacting immune responses and tumor progression [41,42,43].

Inflammation, a hallmark of cancer, plays a pivotal role in elevating hepcidin levels and disrupting iron regulation within the tumor microenvironment (TME) [44]. Pro-inflammatory cytokines, including interleukin-6 (IL-6), tumor necrosis factor-alpha (TNF-α), and interleukin-1 beta (IL-1β), are central to this process [45,46]. IL-6, produced by tumor cells and surrounding stromal and immune cells, is critical in this context [47]. It stimulates hepcidin production, exacerbating iron sequestration and impairing erythropoiesis [48]. TNF-α and IL-1β contribute to chronic inflammation, sustaining elevated hepcidin levels and reinforcing the disruption of iron regulation [49,50]. Hepcidin level has been extensively studied across various cancer types, demonstrating its significance in disease progression and prognosis. In non-small cell lung cancer (NSCLC), increased hepcidin expression was observed in both serum and tumor tissues, correlating with lymph node metastasis and advanced clinical stages [51]. Similarly, serum hepcidin levels were significantly elevated in multiple myeloma compared to healthy controls, highlighting its role in anemia and iron homeostasis disorders commonly associated with the disease [52]. These findings collectively underscore the critical role of hepcidin in cancer biology, linking its elevated levels to tumor aggressiveness, metastasis, and poor clinical outcomes. The consistent correlation of high hepcidin levels with disease progression across multiple cancer types supports its evaluation as a therapeutic target and biomarker for improved cancer management. Additionally, high hepcidin levels contribute to epithelial-to-mesenchymal transition (EMT), a process by which cancer cells gain migratory and invasive properties, further promoting metastasis [53]. Targeting hepcidin or the inflammatory pathways that elevate its levels presents a promising therapeutic approach to address anemia and tumor progression, potentially improving cancer patients’ quality of life and clinical outcomes [54,55].

### 2.1. Effect of Iron Deficiency on the TME and Immune Function

Iron deficiency can significantly impair the immune response [56], crucial for effective cancer treatment. Iron is essential for the proliferation and activation of several immune cells, including T-cells, macrophages, and NK cells, responsible for identifying and eradicating cancer cells [57]. When iron levels are insufficient, the function of these immune cells is compromised [57], indirectly impacting the efficacy of immunotherapies such as ICIs and CAR-T cell therapies (Figure 2).

#### 2.1.1. T-Cells

T-cells, central players in the immune system’s defense against cancer [58], depend highly on iron for proper functioning and activation [59]. Iron is vital in supporting the cellular processes that fuel T-cell activity, particularly during immune responses [60]. When a T-cell encounters a tumor cell, it rapidly proliferates and becomes activated, processes that demand substantial energy [61]. This energy is primarily generated in the mitochondria, where iron is pivotal in ATP production [62]. Without sufficient iron, mitochondrial function becomes impaired, leading to decreased ATP production [62]. This energy deficit directly limits the ability of T-cells to increase, which reduces their ability to respond effectively to tumor cells and other immune challenges.

In addition to its role in energy production, iron also influences gene regulation in T-cells via epigenetic mechanisms [63]. Iron modifies histones, the proteins around which DNA is wrapped, which helps regulate gene expression [63]. In the absence of adequate iron, repressive histone marks become more prevalent, inhibiting the expression of critical genes necessary for T-cell differentiation and function [64]. For example, the differentiation of Th17 cells, a subset of T-cells that plays a crucial role in the immune response, is significantly impaired when iron levels are low [65,66]. Th17 cells are essential for producing cytokines like IL-17, which help orchestrate inflammation and immune defense, particularly against tumors [67]. Without adequate iron, the suppression of Th17 differentiation may weaken the body’s immune defense, making it more challenging to control tumor growth.

Moreover, T-cells require iron to produce cytokines essential for immune activity, such as interferon-gamma (IFN-γ) [68]. IFN-γ enhances the immune system’s ability to attack and destroy cancer cells [69]. This cytokine is critical in driving the cytotoxic activity of T-cells, enabling them to identify and eliminate tumor cells effectively. When iron levels are deficient, the production of IFN-γ may be reduced, weakening the T-cell’s ability to mount an adequate anti-tumor response. This reduced capacity becomes particularly problematic in immunotherapies such as ICIs, designed to boost the immune system’s ability to target tumors. If T-cells are not functioning optimally due to iron deficiency, their response to these therapies may be weakened, diminishing the overall effectiveness of cancer treatment. Thus, iron deficiency has a broad and profound impact on T-cells. It disrupts their energy metabolism, hinders essential gene expression required for differentiation, and limits the production of critical cytokines vital for a robust immune response. These impairments can lead to the reduced control of tumor growth in cancer patients and contribute to the suboptimal outcomes of immunotherapies.

#### 2.1.2. Macrophages

Macrophage polarization significantly influences iron metabolism, with M1 and M2 macrophages exhibiting distinct iron-handling properties [70]. M1 macrophages, characterized by their pro-inflammatory activity, are typically activated by stimuli such as interferon-gamma (IFN-γ) TNF-α, granulocyte-macrophage colony-stimulating factor (GM-CSF), and lipopolysaccharides (LPS) [71]. These signals drive M1 macrophages to sequester iron intracellularly, limiting its availability to pathogens and tumor cells [72]. This is achieved through the low expression of ferroportin (SLC40A1), the primary iron exporter, and upregulation of NRAMP1, which transports iron into lysosomes for storage or utilization [72].

Conversely, M2 macrophages, which play a role in tissue repair and tumor progression, are polarized by factors such as interleukin-4 (IL-4) and interleukin-13 (IL-13) [73]. These signals promote high levels of ferroportin expression, facilitating the export of iron into the extracellular environment [74]. This iron release into the tumor microenvironment promotes cancer cell proliferation and supports the immunosuppressive conditions characteristic of M2 macrophages. These differences in iron metabolism between M1 and M2 macrophages contribute to the dynamic regulation of iron availability in the tumor microenvironment, influencing both tumor progression and immune responses [75].

M1 macrophages rely heavily on iron to generate ROS [75] (Figure 3). These ROS are essential for the macrophages’ cytotoxic function, allowing them to destroy pathogens and tumor cells [75]. When iron levels are insufficient, the production of ROS is significantly reduced [75,76]. This decrease can weaken the M1 macrophages’ ability to attack and kill cancer cells, compromising the body’s anti-tumor immune response. In contrast, M2 macrophages, activated by IL-4 and IL-13, contribute significantly to tumor growth, metastasis, and immune evasion (Figure 3). They promote angiogenesis by secreting factors such as VEGF, which supports tumor proliferation by supplying blood and nutrients [77].

Additionally, M2 macrophages secrete IL-10 and TGF-β, creating an immunosuppressive environment that weakens the immune system’s ability to attack cancer cells [77]. This allows tumors to evade immune detection and spread. M2 macrophages also remodel the extracellular matrix, facilitating cancer cell migration and metastasis [77]. Furthermore, macrophages in the TME play a pivotal role in regulating the immune response through the secretion of various cytokines, including TNF-α, IL-6, and IL-1β [77]. Each of these cytokines has distinct effects on tumor progression and immune modulation.

TNF-α is a crucial cytokine produced by tumor-associated macrophages (TAMs), with dual roles in the TME [77,78]. When secreted by M1 macrophages, TNF-α stimulates an inflammatory response, generating reactive species such as superoxide radicals that promote tumor cell destruction and anti-tumor immunity [77,78]. However, when produced by M2 macrophages or in chronic inflammation, TNF-α can support tumor progression by fostering angiogenesis, immune suppression, and cancer cell survival [77,78]. Similarly, IL-6, secreted by TAMs and other cells in the TME, acts through various signaling pathways, such as STAT3 and NF-κB, promoting tumor growth, metastasis, and cell survival [79,80]. However, acute IL-6 activation can enhance the immune response by recruiting cytotoxic T-cells to target tumor cells, reflecting its complex role in cancer progression [81]. IL-1β, primarily secreted by myeloid-derived suppressor cells (MDSCs) and macrophages, also has dual functions [82]. While IL-1β produced by MDSCs contributes to immune suppression and tumor growth, IL-1β from dendritic cells (DCs) can stimulate T-cell immunity, enhancing anti-tumor responses [82].

#### 2.1.3. NK Cells

NK cells are another critical immune system component directly affected by iron deficiency [83]. NK cells kill tumor cells by releasing cytotoxic molecules such as perforin and granzyme [66]. These molecules create pores in the membrane of cancer cells, allowing the entry of enzymes that induce apoptosis or programmed cell death [84].

Iron plays a critical role in maintaining the cytotoxic function of NK cells. When iron levels are insufficient, the cytotoxic activity of NK cells is significantly reduced [83]. Iron is essential for NK cells’ proper activation and function. Without adequate iron, NK cells struggle to produce the necessary cytotoxic molecules, impairing their ability to eliminate tumor cells effectively [83]. This reduction in NK cell activity is particularly concerning in cancer, where a robust innate immune response is essential for controlling tumor growth.

Additionally, iron deficiency impairs the production of IFN-γ, a key cytokine produced by NK cells that stimulates other immune cells and enhances tumor cell destruction [85]. The reduced output of IFN-γ weakens the overall immune response, further diminishing the ability of NK cells to control tumor growth [86,87]. This is particularly problematic in cancer patients, where NK cell activity is crucial for keeping tumor cells in check and preventing metastasis.

## 3. Iron Overload

### 3.1. Iron Overload in Cancer Patients

Cancer cells can uniquely manipulate iron metabolism, creating an iron-rich microenvironment that supports their malignant phenotype [16]. However, this localized iron overload contrasts with the systemic iron deficiency or anemia often observed in cancer patients, driven by chronic inflammation, malnutrition, or treatment-related factors [88].

This duality complicates the patient’s condition, impacting immune function and overall prognosis.

Tumor cells sequester iron, fueling their growth and exacerbating systemic iron imbalances, potentially worsening patient outcomes [89]. Understanding the intricate relationship between tumor-specific iron deregulation and systemic iron homeostasis is crucial for developing therapeutic strategies that address local and systemic iron metabolism in oncology.

Iron overload plays a critical role in cancer progression, primarily through its impact on cellular metabolism and the induction of oxidative stress [19,20]. When the body accumulates excess iron, this increases ROS production, which are harmful molecules that can cause significant damage to DNA, proteins, and cell membranes [19,20]. This oxidative stress promotes mutations, genetic instability, and crucial tumor development and progression drivers [89,90] (Figure 2). Cancer cells exhibit a highly deregulated iron metabolism that facilitates their growth and survival [89,90]. This deregulation is marked by an increased labile iron pool (LIP), overexpression of transferrin receptors (TfRs), and altered activity of iron regulatory proteins (IRPs), particularly IRP2, which plays a pivotal role in maintaining iron homeostasis within tumor cells [91,92]. These mechanisms create an iron-rich environment that supports key cancer cell processes such as DNA synthesis, oxidative stress response, and metastatic potential.

For instance, in breast cancer and lung cancer, elevated TfR expression has been correlated with increased tumor aggressiveness and poor prognosis [93,94]. Similarly, IRP2 activity has been shown to enhance the iron uptake capacity of melanoma cells, contributing to their invasive phenotype [95,96]. These findings underscore the importance of localized iron overload in driving tumor biology. Cancer cells have a higher iron requirement than normal cells due to their rapid proliferation and increased metabolic needs [97]. They upregulate iron import mechanisms, such as transferrin receptor 1 (TFR1), which allows them to absorb more iron from the bloodstream [97]. However, a cell that stores more iron becomes more susceptible to iron-induced oxidative stress, paradoxically supporting tumor growth. High levels of oxidative stress can even lead to cell death [98].

In patients with genetic predispositions like hereditary hemochromatosis, in which iron regulation is impaired, the risk of cancer is notably higher [99]. These patients often develop hepatocellular carcinoma (HCC) due to chronic iron overload in the liver, which leads to inflammation, fibrosis, and cirrhosis before eventually progressing to cancer [99,100]. In these cases, excess iron promotes continuous oxidative damage, fueling chronic liver disease and creating a fertile environment for cancer initiation [99,100].

Moreover, dietary factors influence iron overload and cancer risk [101]. Heme iron, found in red and processed meats, is absorbed more efficiently by the body than non-heme iron from plant sources [101]. High consumption of heme iron has been associated with an increased risk of cancers, especially colorectal cancer [102]. Heme iron promotes the formation of N-nitroso compounds and free radicals in the gut, leading to DNA damage and increasing the likelihood of cancerous transformations in colon cells [102].

Iron accumulation in cancer cells also supports metabolic changes, making tumors more aggressive [103]. One such change is the Warburg effect, whereby cancer cells rely on glycolysis for energy production, even in the presence of oxygen [103]. Iron is a critical cofactor in glycolysis and DNA synthesis enzymes, allowing cancer cells to sustain their rapid growth [103]. As iron overload increases, the metabolic adaptability of cancer cells improves, giving them a survival advantage in harsh environments such as the low-oxygen (hypoxic) conditions commonly found within tumors [16].

Furthermore, iron overload can make cancer cells more resistant to treatments [104]. Excess ROS from high iron levels can induce mutations that enhance cancer cell survival mechanisms [104]. By improving their antioxidant capacity, iron-overloaded cancer cells become more resilient to these treatments, making the disease more difficult to manage [104]. For example, cancer cells under oxidative stress often activate antioxidant defense systems, such as the glutathione system, to protect themselves from ROS-induced damage [105,106]. This can lead to resistance against therapies like chemotherapy and radiation, which rely on generating oxidative stress to kill cancer cells [105,106].

Recent therapeutic strategies have aimed to target iron metabolism in cancer treatment. Iron chelation therapy, which involves using agents like deferoxamine or deferasirox, can potentially reduce iron availability to cancer cells, limit their growth, and increase their susceptibility to oxidative stress [107]. By binding to excess iron, these chelators prevent iron from participating in harmful reactions that generate ROS, reducing the overall oxidative burden on the body and slowing down cancer progression [108]. In addition to chelation, targeting iron metabolism through pathways like ferroptosis, an iron-dependent form of cell death, is also being explored as a promising therapeutic approach [109].

### 3.2. Effect of Iron Overload on the TEM and Cancer Cells

#### 3.2.1. T-Cells

Iron overload significantly affects T-cells, particularly CD8^+^ cytotoxic T-cells and CD4+ helper T-cells, by impairing their function and expansion [25]. Optimal levels of intracellular iron are crucial for proper IL-2 receptor (IL-2R) signal transduction and mitochondrial function, both critical for T-cell proliferation [110]. Upon activation, T-cells experience a reduction in intracellular iron, leading to an accumulation of iron in the surrounding environment, which indicates that T-cells actively regulate iron levels to avoid overload while responding to stimulation through the T-cell receptor (TCR) [110]. In patients with hereditary hemochromatosis, iron overload leads to an increase in the number and activity of suppressor CD8^+^ T-cells while reducing the proliferation, numbers, and function of CD4+ helper T-cells, resulting in an elevated CD8^+^/CD4+ ratio [111]. This imbalance impairs the generation of cytotoxic T-cells and alters immunoglobulin secretion compared to individuals with treated HH or healthy controls [111]. Moreover, iron overload can further exacerbate immune dysfunction by depleting CD4+ T-lymphocytes by shortening their lifespan and impairing the phagocytic activity of CD8^+^ CD28 T-lymphocytes [112]. This cumulative effect weakens the immune response, increasing susceptibility to infections due to the compromised functionality of key immune cells [112].

This combination of effects on T-cells demonstrates the profound impact of iron overload on immune regulation, particularly in patients with conditions like HH, where iron metabolism is chronically disrupted.

#### 3.2.2. NK Cells

In cases of iron overload, such as in patients with HH, the effect on NK cells can be less detrimental or neutral. Some studies show that NK cells from HH patients maintain normal degranulation (the process of releasing cytotoxic substances) and cytotoxic activity, with no significant differences compared to healthy controls. Interestingly, some HH patients exhibited slightly higher cytotoxic activity, though these differences were not statistically significant [113]. Other findings strongly indicate that NK cell function in peripheral blood is not significantly impaired in patients with HH, suggesting that iron overload does not directly compromise NK cells’ cytotoxic abilities or overall immune performance [114].

## 4. Ferroptosis

Beyond its impact on immune cells, iron overload influences other critical processes in cancer biology, one of the most notable being ferroptosis (Figure 3). Ferroptosis is a form of regulated cell death distinct from different types of cell death, primarily due to its dependence on iron and oxidative stress [115]. Unlike apoptosis, which involves caspase activation and DNA fragmentation or necrosis, an uncontrolled form of cell death, ferroptosis is triggered by the accumulation of intracellular iron and the production of ROS [116]. These ROS target and oxidize polyunsaturated fatty acids in the cell membrane, causing lipid peroxidation [117]. As the peroxidized lipids accumulate, the structural integrity of the cell membrane deteriorates, ultimately leading to cell death [118].

The fundamental difference between ferroptosis and other forms of cell death lies in its unique dependency on iron and lipid peroxidation [119]. Apoptosis is primarily regulated through protein cleavage and DNA fragmentation via caspases [120], while necrosis involves a loss of membrane integrity without the regulated pathways seen in ferroptosis [121]. Autophagy, another regulated process, degrades cellular components through lysosomes [122]. Ferroptosis, by contrast, is centered on iron-mediated oxidative damage to cellular membranes [123].

In cancer biology, ferroptosis plays a vital role [124]. Cancer cells, especially those with altered iron metabolism, are highly susceptible to ferroptosis due to their reliance on iron for rapid growth and proliferation [125]. Activating ferroptosis in cancer cells can be a therapeutic mechanism to limit tumor growth [126]. Additionally, ferroptosis induction can trigger immune cell responses [127]. When cancer cells undergo ferroptosis, they release damage-associated molecular patterns (DAMPs) recognized by immune cells such as DCs, macrophages, and T-cells [128]. This immune activation enhances anti-tumor immunity by recruiting immune cells to the TME and promoting the clearance of cancer cells [129].

Thus, ferroptosis can be critical in tumor and immune cells. Iron overload sensitizes tumor cells to ferroptosis, which can be exploited therapeutically to induce selective cancer cell death [130,131]. For instance, cancer cells with high levels of labile iron pools are particularly susceptible to ferroptosis, highlighting its relevance as a potential treatment strategy [130,131].

Conversely, in immune cells, ferroptosis may disrupt anti-tumor immunity by inducing cell death in key immunocompetent cells, such as T-cells or macrophages [132]. This dual role underscores the complexity of ferroptosis in the tumor microenvironment. Further exploration of ferroptosis, particularly its regulation by iron metabolism, could reveal new therapeutic approaches to target tumor cells while preserving immune function.

Combining ferroptosis-inducing therapies with immunotherapies, such as ICIs, could lead to synergistic effects, where the cancer is attacked on multiple fronts through direct cell death and enhanced immune responses. However, T-cells lacking GPX4 rapidly accumulate membrane lipid peroxides, leading to ferroptosis and impairing their ability to mount an effective immune response, highlighting the critical role of GPX4 in T-cell immunity [133]. Similarly, ACSL4 is essential for the ferroptosis of CD8^+^ T-cells and their immune functions [134].

## 5. Clinical and Preclinical Evidence

### 5.1. Clinical and Preclinical Evidence of Iron Deficiency

A growing body of clinical and preclinical studies provides evidence of how iron deficiency impacts cancer progression, immune function, and treatment efficacy. In preclinical models, iron deficiency has been shown to impair the immune response, particularly in the activation and function of T-cells. For example, studies in murine models have demonstrated that iron deficiency reduces T-cell proliferation, cytokine production, and the ability to combat tumor cells effectively [26]. These findings align with clinical observations where iron deficiency in cancer patients is associated with weakened immune responses, reduced treatment tolerance, and poor overall outcomes [25].

For cancer patients, iron deficiency leads to anemia and impairs the efficacy of immune-based treatments, such as ICIs and CAR-T cell therapies [16]. Research shows that patients with iron deficiency often exhibit lower response rates to immunotherapies. This is particularly critical for treatments like ICIs, which rely on a robust immune response to target and eliminate cancer cells [20]. Moreover, iron deficiency has been linked to increased fatigue, poorer quality of life, and reduced physical performance, all of which further impact cancer treatment outcomes [14].

Clinical studies have explored iron supplementation as a potential strategy to improve treatment outcomes in iron-deficient cancer patients. Iron replacement therapy has been shown to restore hemoglobin levels and alleviate symptoms of anemia, but there is a growing focus on how iron supplementation may also enhance immune responses [26]. Some studies suggest that correcting iron deficiency could help improve the effectiveness of ICIs by bolstering the immune system’s ability to respond to tumors [24]. However, the risk of iron overload with supplementation highlights the need for careful monitoring in cancer patients, emphasizing the importance of personalized management strategies.

### 5.2. Clinical and Preclinical Evidence Regarding Iron Overload

Iron overload has been shown to promote tumor growth and reduce the effectiveness of cancer therapies, both in clinical and preclinical settings. Deregulations in iron metabolism are observed across various tumor types and are critical to cancer progression. For example, breast cancer cells show upregulated hepcidin expression, suppressing ferroportin activity and leading to iron accumulation that supports tumor growth [135]. Blocking IRP2-mediated alterations in iron metabolism effectively hinders tumor growth in colorectal cancer [136]. Glioblastoma cells sequester iron, enabling survival in hypoxic conditions [137,138]. These examples underscore the importance of understanding tumor-specific alterations in iron metabolism to identify potential therapeutic targets. Preclinical studies in mouse models demonstrate that iron overload enhances tumor progression by generating ROS, which promote genetic mutations and fuel cancer cell proliferation [19,20]. Iron overload was also found to impair immune responses in these models, mainly by altering the function of CD8^+^ T-cells and promoting immunosuppressive M2 macrophage polarization [75].

Iron overload significantly increases the risk of developing cancers such as HCC in patients with HH or those receiving frequent blood transfusions [99]. The chronic deposition of iron in the liver leads to inflammation, fibrosis, and cirrhosis, creating an environment conducive to tumor development [99,100]. Additionally, clinical studies have demonstrated that patients with high iron levels exhibit poorer responses to chemotherapy and immunotherapy, as their tumors are more resistant to treatments that rely on oxidative stress to kill cancer cells [104].

Despite these risks, some studies suggest higher serum iron levels improve responses to ICIs in specific patient populations. For example, a survey of patients with advanced metastatic cancer found that those with higher serum iron levels (>1036 μg/L) showed better treatment responses to ICIs than patients with lower iron levels [139]. This unexpected finding points to the complex role of iron in cancer therapy, where both deficiency and overload can have opposing effects depending on the context.

### 5.3. Therapeutic Strategies and Future Directions

The dual role of iron in cancer—where deficiency impairs immune function and treatment efficacy while overload promotes tumor progression—has spurred interest in therapeutic strategies targeting iron metabolism. Two key approaches, iron chelation therapy and ferroptosis induction, have shown promise in preclinical and emerging clinical studies.

Iron chelation therapy aims to reduce excess iron levels in cancer patients, depriving tumor cells of this critical resource for their growth and proliferation [140]. Agents like deferoxamine (DFO) and deferasirox (DFX) have been extensively studied in preclinical settings [141]. Deferasirox (DFX) has been demonstrated to enhance the effectiveness of chemotherapy in triple-negative breast cancer (TNBC) by suppressing cell proliferation, promoting apoptosis and autophagy, and delaying tumor recurrence, all without increasing treatment-related toxicity [142]. TSC24, a potent iron chelator, has been shown to effectively inhibit human HCC tumor growth in studies by disrupting iron homeostasis, depleting available iron, and inducing cell-cycle arrest and apoptosis, with minimal toxicity observed at therapeutic doses, highlighting its potential as a treatment for HCC [143].

Clinical applications of iron chelation therapy have also been explored. For example, a pilot study involving patients with advanced HCC demonstrated that DFO treatment reduced serum alpha-fetoprotein (AFP) levels, a marker of tumor activity, suggesting potential anti-tumor effects [144]. Similarly, adding DFO to standard chemotherapy improved survival outcomes in a small cohort of patients with leukemia, likely due to the chelation of iron essential for leukemic cell growth [145].

Ferroptosis has emerged as a novel therapeutic target in cancer treatment. Preclinical research highlights its potential to selectively kill tumor cells by overwhelming their antioxidant defenses and inducing lethal lipid peroxidation [146]. Agents such as erastin and RSL3 have been shown to induce ferroptosis in various cancer models [147]. In studies on pancreatic cancer, ferroptosis inducers have been shown to significantly reduce tumor size and improve survival in mouse models by exploiting the high iron dependency of cancer cells [148]. In vitro experiments on glioblastoma U87 and U251 cell lines using a combination of erastin and ionizing radiation (IR) demonstrated a connection between enhanced radiosensitivity and ferroptosis induction [149].

Emerging evidence suggests combining ferroptosis-inducing therapies with immune checkpoint inhibitors (ICIs) could further enhance anti-tumor efficacy. PD-1 membrane-coated RSL3 nanoparticles (PD-1@RSL3 NPs) were shown in preclinical models to enhance breast cancer treatment by inducing tumor ferroptosis, disrupting the PD-1/PD-L1 axis, and activating antitumor immunity, leading to delayed tumor progression and improved survival [150,151]. These findings pave the way for integrating ferroptosis-targeted therapies into combination regimens with immunotherapy.

Despite these promising results, challenges remain. Striking the right balance between iron depletion and maintaining adequate levels for normal physiological functions is critical. Additionally, clinical trials are necessary to evaluate these therapies’ safety, efficacy, and optimal dosing in cancer patients.

Future directions should focus on identifying reliable biomarkers to guide iron-targeted therapies. For instance, measuring transferrin receptor (TfR) expression, ferritin levels, or lipid peroxidation markers could help stratify patients for specific treatments. Tailoring strategies to individual patients allows oncologists to exploit tumor-specific vulnerabilities better while minimizing adverse effects. These advancements can potentially revolutionize personalized cancer treatment and improve patient outcomes.

## 6. Conclusions

This narrative review has explored iron’s dual role in cancer progression, immune function, and treatment efficacy. Iron deficiency in cancer patients can impair immune responses, weaken the effectiveness of ICIs and CAR-T cell treatments, and exacerbate symptoms like fatigue and anemia. On the other hand, iron overload promotes tumor growth, induces oxidative stress, and reduces the effectiveness of treatments like chemotherapy and immunotherapy. Both iron deficiency and overload present significant challenges in oncology, underscoring the need for personalized management of iron levels in cancer care. Clinical and preclinical studies provide valuable insights into the complex relationship between iron metabolism and cancer. Research shows that correcting iron deficiency can improve immune function and boost treatment responses. In contrast, therapeutic strategies targeting iron overload, such as iron chelation and ferroptosis induction, offer promising avenues for reducing tumor growth. However, the exact impact of iron supplementation and overload on cancer treatment remains an area of ongoing investigation, particularly in patients receiving immunotherapy.

Future research should focus on developing more sophisticated biomarkers for iron status, exploring the interplay between iron metabolism and the immune system, and evaluating the therapeutic potential of targeting iron in cancer treatment. By integrating iron management into cancer therapy protocols, oncologists can optimize treatment efficacy and improve outcomes for cancer patients.

## Figures and Tables

**Figure 1 cancers-16-04156-f001:**
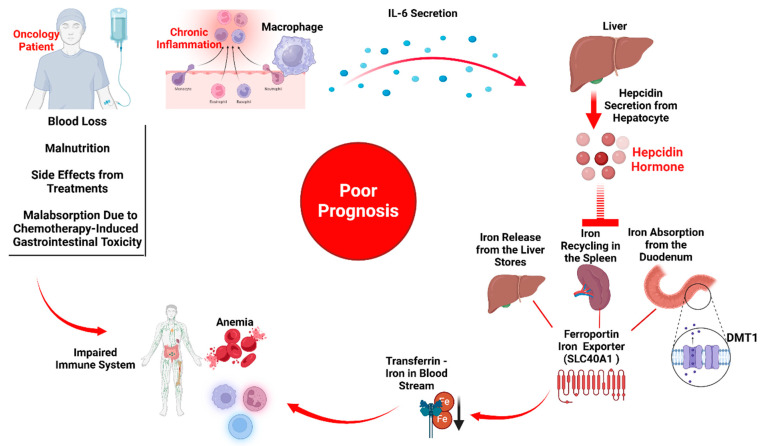
Iron Deficiency in Oncology Patients: Oncology patients may suffer from iron deficiency due to multiple causes, e.g., blood loss, malnutrition, side effects from treatments, malabsorption, and chronic inflammation. Consistent with chronic inflammation, immune cells in the tumor microenvironment (TME), such as macrophages, and secreted pro-inflammatory cytokines, such as IL-6, can trigger hepatocytes to secrete hepcidin (a hormone that regulates iron absorption and distribution). Hepcidin regulates iron levels by blocking the release of iron from liver stores, preventing the recycling of iron from aged red blood cells by macrophages, and reducing iron absorption from the duodenum by affecting divalent metal transporter 1 (DMT1), which is expressed in the absorptive enterocytes, and the degradation of ferroportin (SCL40A1), which is the iron exporter expressed in iron-storing and iron-transporting tissues, i.e., absorptive enterocytes, macrophages, hepatocytes, and placental cells. This condition, functional iron deficiency, correlates with a poor prognosis in oncology patients. The figures were created using BioRender.com.

**Figure 2 cancers-16-04156-f002:**
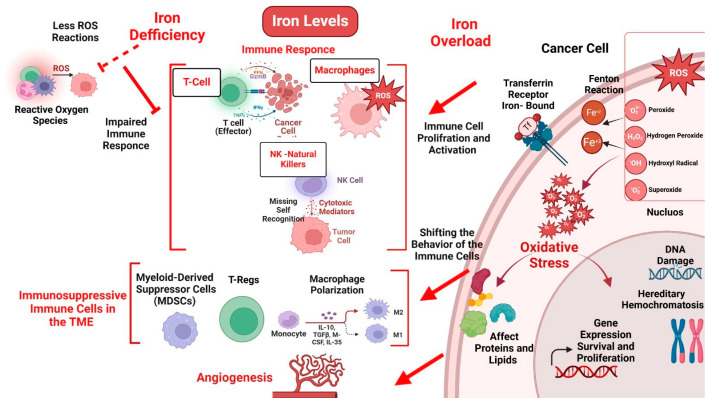
The Effect of Iron Levels on Cancer and Immune Cells: Iron overload can occur in oncology patients due to frequent blood transfusions, excessive iron supplementation, or genetic conditions such as hereditary hemochromatosis. High iron levels can promote the generation of ROS through the Fenton reaction, leading to oxidative stress, DNA damage, and genomic instability. These effects can enhance tumor growth and metastasis by creating a pro-tumorigenic environment.

**Figure 3 cancers-16-04156-f003:**
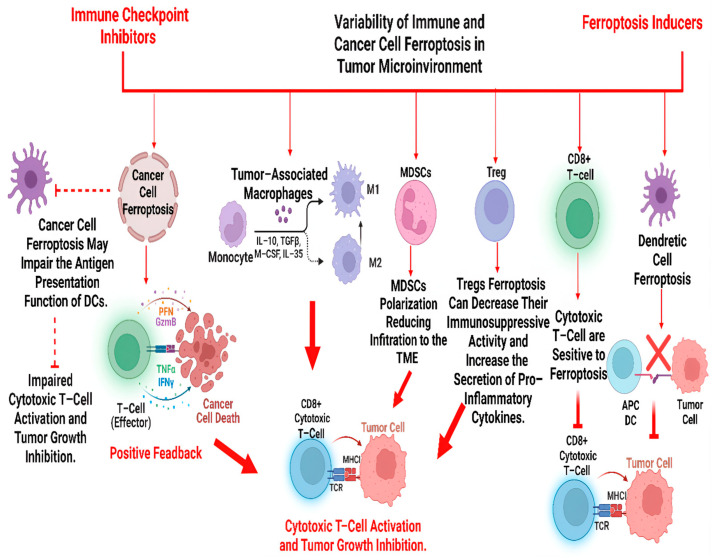
The varying impact of ferroptosis on cancer and immune cells within the TME. Cancer Cell Ferroptosis: Inducing ferroptosis in cancer cells can impair the antigen presentation function of dendritic cells (DCs), leading to impaired activation of cytotoxic T-cells and inhibiting tumor growth. Positive Feedback Loop: The death of cancer cells through ferroptosis can stimulate effector T-cells (CD8^+^ T-cells) by releasing pro-inflammatory cytokines such as TNFα and IFNγ, which further promote anti-tumor responses. Tumor-associated macrophages (TAMs) can polarize into M1 or M2 phenotypes. M1 macrophages are pro-inflammatory and anti-tumorigenic, promoting cancer cell death, while M2 macrophages support tumor growth by secreting immunosuppressive cytokines such as IL-10, TGFβ, M-CSF, and IL-35. Myeloid-derived suppressor cells (MDSCs) contribute to immune evasion by reducing infiltration to the TME and promoting immunosuppression. Ferroptosis in MDSCs can reduce their immunosuppressive activity, potentially enhancing anti-tumor immunity. Treg ferroptosis can decrease their immunosuppressive activity and increase the secretion of pro-inflammatory cytokines. CD8^+^ T-cells are sensitive to ferroptosis. Excess iron can inhibit their activation, expansion, and survival, reducing the efficacy of immune responses against tumor cells. Ferroptosis in DCs can impair their function in antigen presentation, leading to reduced activation of cytotoxic T-cells and decreased tumor growth inhibition.

## Data Availability

No new data were created in the writing of this review.

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
