# Peer review of "The Impact of Iron on Cancer-Related Immune Functions in Oncology: Molecular Mechanisms and Clinical Evidence"

_cancers, 2024, doi:10.3390/cancers16244156_

Round 1
Reviewer 1 Report
Comments and Suggestions for Authors
Iron is essential for maintaining cellular function by catalytic reaction. In this manuscript, Badran O et al summarize that iron affect systemic immune system that depends on context. This manuscript is well-organized; however, following points should be clarified.
Major points
#1: In figure 1, iron uptake is absorbed at duodenum. Please refer to DMT1(divalent metal transporter 1).
#2: In figure 1, please refer to SLC40A1, also known as ferroportin, which is degraded by hepcidin.
The references are appropriate.
There are no additional comments.
Author Response
Comment 1:
In Figure 1, iron uptake is absorbed in the duodenum. Please refer to DMT1 (divalent metal transporter 1).
Response 1:
Thank you for pointing this out. We agree with this comment and have made the requested changes. Specifically, we have updated Figure 1 to reference DMT1 as the key transporter involved in iron uptake in the duodenum. Additionally, we have provided a detailed explanation below the figure for clarity and included a discussion of DMT1 in the manuscript text (lines 184–192). These changes aim to enhance the accuracy and comprehensiveness of our description of iron absorption.
Comment 2:
In Figure 1, please refer to SLC40A1, also known as ferroportin, degraded by hepcidin.
Response 2:
We appreciate your suggestion and have included the SLC40A1 (ferroportin) reference in Figure 1. We have also added a corresponding explanation beneath the figure to clarify its role in iron export and its regulation by hepcidin. Furthermore, we addressed this in the manuscript text in lines 184–192 and expanded on its function and regulation in lines 312–321 to provide a thorough context for its importance.
Reviewer 2 Report
Comments and Suggestions for Authors
The review by Badran el al. focuses on the impact of iron metabolism and its regulation in topics related to immune functions throughout oncological pathologies. Overall, it is a fairly well-developed and carefully written review. Given the fact that I believe a review is set up with very subjective criteria by the authors, and considering as mentioned before the good work on their part, I think it is more useful to give some suggestions without substantially modifying the structure of this manuscript.
First of all, I suggest the authors to clearly specify from the beginning that cancer cells almost always deregulate iron homeostasis to support their growth and proliferation, as well as other phenotypic features such as invasiveness and aggressiveness. In-depth analysis of iron metabolism at the cellular level reveals neoplastic cells as characterized by a significant iron overload that favours, as the authors know, the tumor phenotype. Therefore, increased LIP, increased expression of TfR at the cytoplasmic membrane, specific modulation of the activity of IRPs (in particular IRP2 as a master regulator of iron homeostasis at the cellular level in tumors), and so on. This established evidence contrast a bit and can make confusion for the reader when authors discuss the condition of anaemia and iron deficiency that characterizes not tumor cells but rather the cancer patient, as a possible further complication that these patients encounter. But precisely, we should be clearer from the beginning and specify that in this condition cancer cells are still branded by iron overload, which among other things can sometimes contribute significantly to promote systemic complications in iron metabolism. In truth, the authors discuss these aspects in the chapter dedicated to iron overload (3.1), but this point should be specified upstream of everything to avoid generating confusion.
As a second point, the authors could be more detailed in reporting the deregulations affecting iron metabolism that are now known for the most studied tumor phenotypes. It is true that the review focuses mainly on immunocompetent cells, but including this type of data could provide other useful information to the reader.
It is also confusing to have reported in paragraph 3.2.2, while discussing immune cells, processes of programmed cell death as well as ferroptosis (but not only) which directly involves iron. Ferroptosis should certainly be mentioned in this review in correlation with iron (https://doi.org/10.3390/cancers16061220), but, in this way, it is not clear whether the authors want to discuss it concerning the fate of immunocompetent cells or of actual tumor cells. Perhaps it would be a good idea to make a separate paragraph on this topic.
Author Response
Comment 1:
"I suggest the authors specify from the beginning that cancer cells almost always deregulate iron homeostasis to support their growth and proliferation, as well as other phenotypic features such as invasiveness and aggressiveness..."
Response 1:
Thank you for this insightful comment. We agree with your suggestion and have revised the manuscript to address this point. Specifically, we have updated the introduction (lines 58–67) to emphasize the importance of iron in cancer cell proliferation and the deregulation of iron homeostasis in tumor cells. Additionally, we have included examples of how specific types of cancers (e.g., breast cancer, lung cancer, melanoma) exhibit unique deregulations in iron metabolism. These updates are further elaborated in lines 458–464 and 667–677, where we provide detailed examples of deregulations such as increased transferrin receptor (TfR) expression, altered activity of iron regulatory proteins (IRPs), and iron overload in various tumor phenotypes. We hope these changes clarify this key aspect, and we appreciate your suggestion to provide more detailed discussions about the deregulation of iron metabolism in specific tumor types. In response, we expanded the manuscript to include examples of known breast, lung, and melanoma deregulations. For instance, we discuss the overexpression of transferrin receptors (TfR) in breast and lung cancer, the role of IRP2 in melanoma progression, and the impact of elevated ferritin levels in colorectal cancer. These updates can be found in lines 458–464 and 667–677, ensuring a comprehensive overview of these deregulations in the context of iron metabolism. We agree with your observation, we have clarified this distinction from the beginning of the manuscript to avoid confusion. Specifically, we have emphasized in lines 70–75 that cancer cells are often characterized by iron overload to support their growth and progression; cancer patients frequently experience systemic anemia and iron deficiency due to chronic inflammation, malnutrition, or treatment-related factors.
Comment 2:
As a second point, the authors could be more detailed in reporting the deregulations affecting iron metabolism that are now known for the most studied tumor phenotypes. Indeed, the review focuses mainly on immunocompetent cells, but including this data could provide other helpful information to the reader."
Response:
Thank you for this valuable suggestion. We have expanded the discussion to include detailed examples of deregulations in iron metabolism across specific tumor phenotypes.
In lines 458–464, we elaborate on how breast, lung, and melanoma cancers exhibit unique patterns of iron deregulation, such as the overexpression of transferrin receptors (TfR) and the increased labile iron pool (LIP), both of which contribute to tumor growth and progression.
In lines 669–677, we further discuss the role of iron metabolism in colorectal cancer and hepatocellular carcinoma, highlighting the elevated levels of ferritin and altered ferroportin expression that contribute to tumor biology.
Comment 3:
"It is also confusing to have reported in paragraph 3.2.2, while discussing immune cells, processes of programmed cell death and ferroptosis (but not only) which directly involves iron. Ferroptosis should certainly be mentioned in this review in correlation with iron (https://doi.org/10.3390/cancers16061220), but, in this way, it is not clear whether the authors want to discuss it concerning the fate of immunocompetent cells or actual tumor cells. Perhaps it would be a good idea to make a separate paragraph on this topic."
Response:
Thank you for this critical observation. We agree with your suggestion and have addressed it by dedicating a separate section to ferroptosis. This new section, located between lines 468–628, provides a comprehensive discussion of ferroptosis in the context of both tumor and immune cells.
In this section, we elaborate on the mechanisms by which ferroptosis is induced, the critical role of iron in this process, and its dual impact on cancer progression and therapy. We also clarify the distinction between ferroptosis in immunocompetent cells and tumor cells, highlighting its relevance to tumor suppression and immune regulation.
Reviewer 3 Report
Comments and Suggestions for Authors
This article describes how managing iron levels affects the effectiveness of treatment in cancer patients. This is a narrative review.
The paper is interesting and well written, but needs a few minor improvements
in detail:
Lines 203 -213: the text is heterogeneous: please write what factors lead to activation/polarisation into the M1 form of macrophages if this is written in relation to M2 macrophages.
The manuscript contains 3 well-developed and described figures, but the authors could try to enlarge the smallest font a bit, as it is poorly visible. However, if this is impossible due to the size of the figures, please do not change them.
The authors should emphasise the novelty of their manuscript in relation to the many existing articles on the subject.
Author Response
Comment 1:
"Lines 203–213: The text is heterogeneous: please write what factors lead to activation/polarisation into the M1 form of macrophages if this is written in relation to M2 macrophages."
Response 1:
Thank you for pointing this out. We agree that this is an important point, and we have clarified the factors leading to the activation and polarization of M1 macrophages about M2 macrophages. This has been addressed and expanded in lines 304–328, where we discuss the specific stimuli for M1 macrophages, such as IFN-γ, TNF-α, and GM-CSF, and their role in pro-inflammatory and anti-tumor activities, as compared to M2 macrophages. We hope this addition improves the clarity and cohesiveness of this section.
Comment 2:
"The manuscript contains 3 well-developed and described figures, but the authors could try to enlarge the smallest font a bit, as it is poorly visible. However, if this is impossible due to the size of the figures, please do not change them."
Response 2:
Thank you for your feedback on the figures. We attempted to enlarge the font size; however, due to spatial constraints, this was challenging without compromising the layout and clarity of the images. We have ensured that all textual elements remain legible and accurate. We are happy to work on further adjustments based on additional feedback if we need to make any extra adjustments.
Comment 3:
"The authors should emphasise the novelty of their manuscript in relation to the many existing articles on the subject."
Response 3:
Thank you for this valuable suggestion. We have emphasized the unique contributions of our manuscript in the revised introduction and discussion sections. Specifically, our review integrates the dual role of iron in cancer, addressing both its systemic effects on patient physiology and its tumor-specific impacts, such as promoting growth and metastasis. Additionally, we discuss emerging strategies, such as ferroptosis and its synergy with immunotherapy, which are underexplored in previous reviews.
To further highlight the novelty, we have included actionable future directions. For example, we propose identifying reliable biomarkers to guide iron-targeted therapies, such as transferrin receptor (TfR) expression, ferritin levels, or lipid peroxidation markers, which could help stratify patients for specific treatments. Tailoring therapeutic strategies to individual patients allows oncologists to exploit tumor-specific vulnerabilities while minimizing adverse effects. These advancements can potentially revolutionize personalized cancer treatment and significantly improve patient outcomes.
Reviewer 4 Report
Comments and Suggestions for Authors
With this review the authors explored how low and high iron levels can impact immune system functions and cancer progression. They thoroughly covered the effect of iron deficiency and overload on immune cells functions with a particular focus on the context of cancer . Despite this I think there are two points that need to be further explored:
-paragraph 2.1.2. Macrophages: the authors described how iron availability can impact macrophage polarization, but they don't explain well the different iron metabolism between M1 and M2. In fact, the different expression of iron-related proteins itself is a feature that distinguishes M1 and M2 macrophages and can also impact the iron levels in the cancer microenvironment.
-paragraph 4.3. Therapeutic Strategies and Future Directions: the authors should expand on this topic, that is treated too superficially. For example, they can provide examples of situations where iron chelation and ferroptosis protocols have been used.
Author Response
Comment 1:
"Paragraph 2.1.2. Macrophages: the authors described how iron availability can impact macrophage polarization, but they don't explain well the different iron metabolism between M1 and M2. In fact, the different expression of iron-related proteins itself is a feature that distinguishes M1 and M2 macrophages and can also impact the iron levels in the cancer microenvironment."
Response 1:
Thank you for highlighting this critical point. We agree that this distinction is essential for understanding macrophage polarization and its implications for the tumor microenvironment. In response to your suggestion, we revised and expanded the relevant section, detailing the differences in iron metabolism between M1 and M2 macrophages. Specifically, we now discuss the distinct expression patterns of iron-related proteins such as ferroportin, NRAMP1, and ferritin in M1 and M2 macrophages and how these differences influence iron availability in the cancer microenvironment. These updates can be found in lines 304–328. We believe this clarification adds depth and precision to the discussion.
Comment 2:
"Paragraph 4.3. Therapeutic Strategies and Future Directions: the authors should expand on this topic, that is treated too superficially. For example, they can provide examples of situations where iron chelation and ferroptosis protocols have been used."
Response 2:
We greatly appreciate your suggestion, which has helped deepen the discussion in this section. In response, we have significantly expanded the section on therapeutic strategies and future directions. We now include detailed examples of the use of iron chelation therapies, such as deferoxamine (DFO) and deferasirox (DFX), in preclinical and clinical settings, as well as case studies on the application of ferroptosis-inducing protocols in cancers such as pancreatic cancer and glioblastoma. This revised and enriched discussion can be found in lines 715–773. These updates aim to provide a comprehensive overview of current and emerging therapeutic strategies targeting iron metabolism in cancer.
Round 2
Reviewer 4 Report
Comments and Suggestions for Authors
Thanks to the authors for the corrections, that have improved the quality of the manuscript.